# Genomic Investigation of *Salmonella* Typhi in Hong Kong Revealing the Predominance of Genotype 3.2.2 and the First Case of an Extensively Drug-Resistant H58 Genotype

**DOI:** 10.3390/microorganisms11030667

**Published:** 2023-03-06

**Authors:** Xin Li, Huiluo Cao, Jonathan Hon-Kwan Chen, Yuey-Zhun Ng, Ka-Kin Fung, Vincent Chi-Chung Cheng, Pak-Leung Ho

**Affiliations:** 1Department of Microbiology, and Carol Yu Centre for Infection, School of Clinical Medicine, Li Ka Shing Faculty of Medicine, The University of Hong Kong, Pokfulam, Hong Kong SAR, China; 2Department of Microbiology, Queen Mary Hospital, Hong Kong SAR, China; 3Department of Clinical Pathology, Pamela Youde Nethersole Eastern Hospital, Hong Kong SAR, China

**Keywords:** *Salmonella* Typhi, typhoid fever, XDR, H58, Hong Kong

## Abstract

Typhoid fever is a notable disease in Hong Kong. We noticed two local cases of typhoid fever caused by *Salmonella* Typhi within a two-week period in late 2022, which had no apparent epidemiological linkage except for residing in the same region of Hong Kong. A phylogenetic study of *Salmonella* Typhi isolates from Hong Kong Island from 2020 to 2022 was performed, including a whole-genome analysis, the typing of plasmids, and the analysis of antibiotic-resistance genes (ARGs), to identify the dominant circulating strain and the spread of ARGs. A total of seven isolates, from six local cases and an imported case, were identified from positive blood cultures in two hospitals in Hong Kong. Five antibiotic-sensitive strains of genotype 3.2.2 were found, which clustered with another 30 strains originating from Southeast Asia. Whole-genome sequencing revealed clonal transmission between the two index cases. The remaining two local cases belong to genotype 2.3.4 and genotype 4.3.1.1.P1 (also known as the H58 lineage). The genotype 4.3.1.1.P1 strain has an extensively drug-resistant (XDR) phenotype (co-resistance to ampicillin, chloramphenicol, ceftriaxone, ciprofloxacin, and co-trimoxazole). Although the majority of local strains belong to the non-H58 genotype 3.2.2 with a low degree of antibiotic resistance, the introduction of XDR strains with the global dissemination of the H58 lineage remains a concern.

## 1. Introduction

*Salmonella enterica* subsp. *enterica* serotype Typhi (*Salmonella* Typhi) is the causative agent of typhoid fever, a systemic infection with an annual case number of around 10 million globally [1]. It is an exclusively human pathogen that is transmitted via the faecal–oral route, predominantly foodborne from the consumption of foods or water contaminated with faeces from a human case or an asymptomatic carrier [2]. The infective dose has been estimated to be around 10,000 organisms in human challenge studies [3]. Patients usually present after an incubation period of 7 to 14 days (range, 3 to 60) with fever, headache, malaise, myalgia, and a varying combination of enteric, respiratory, neurologic, and dermatologic symptoms. Complications occur in 10–15% of patients, such as gastrointestinal bleeding, intestinal perforation, and typhoid encephalopathy, which can lead to significant mortality and morbidity [2]. With the availability of medical care and antimicrobial treatment, the global case fatality rate in 2017 was estimated to be 0.95%, but mortality rates were still high among people at the extremes of age and those living in lower-income countries [1]. 

*Salmonella* Typhi belongs to subspecies 1 of the broader species *S. enterica*. Under the Kaufmann–White scheme, the antigen formula for *Salmonella* Typhi is group 9 and 12 for O antigen, *d* for H antigen, and *vi* for capsular antigen [3,4]. An alternative phase flagellar II antigen was described for some strains carrying the linear plasmid, pBSSB1 [5]. The high pathogenicity of *Salmonella* Typhi is contributed by the presence of unique *Salmonella* pathogenicity islands (SPIs) encoding the Via locus (expression of the Vi capsule), a bacterial type III secretion system (T3SS), and a type IVB pili-mediating cell attachment [3,4]. A unique feature of typhoid fever is the carrier state, in which *Salmonella* Typhi can be shed in stool for a prolonged time without clinically apparent disease. The pathogenic mechanisms underlying the establishment of the carrier state are poorly understood, but the colonisation of the gallbladder with biofilm formation has been suggested to be the primary cause [3].

The effectiveness of antimicrobial therapy in reducing the mortality and morbidity of typhoid fever is facing increasing threat due to antimicrobial resistance. Multidrug-resistant (MDR) *Salmonella* Typhi, defined as resistant to ampicillin, chloramphenicol, and trimethoprim–sulfamethoxazole (co-trimoxazole), first emerged in the 1970s [6,7] and has displaced drug-sensitive strains in many regions, largely driven by IncHI1 plasmid transfer [8]. Resistance to fluoroquinolones increased since the 1990s, predominantly caused by the chromosomal S83F substitution in the *gyrA* gene [9], which was first identified in Southeast Asia with subsequent intercontinental spread to Africa [10,11]. 

Phylogenetic analysis suggested that the global *Salmonella* Typhi population is highly monophyletic, with an evolutionary process characterised by the ongoing loss of gene function [12]. Recombination events in this serovar were relatively infrequent [12,13]. One widely adopted scheme based on the biallelic profiles of 88 SNPs within a limited set of genes classified the *Salmonella* Typhi population into 85 haplotypes (H1 to H85 genotypes or lineages) [13,14]. An MDR lineage, H58, has become globally dominant with multiple inter- and intra-continental transmission events [15]. In 2016, the first large-scale outbreak of extensively drug-resistant (XDR) *Salmonella* Typhi, defined as MDR isolates harbouring additional resistance to fluoroquinolones and third-generation cephalosporins, occurred in Pakistan [16]. The propensity for H58 isolates to acquire drug resistance is attributed to a cluster of transposon-associated resistance genes residing either on transmissible IncHI1 plasmids or on the chromosome following integration [15]. In 2016, WGS data from a global collection of ~2000 *Salmonella* Typhi isolates was used to develop another SNP typing scheme (GenoTyphi) [17], which used 68 marker SNPs to define 4 primary clades, 16 clades, and 49 subclades in pseudo-hierarchical nomenclature. The H58 lineage was designated as a subclade, called genotype 4.3.1 in the GenoTyphi scheme [18].

With the post-pandemic travelling boom, there is a high risk of the global dissemination of drug-resistant *Salmonella* Typhi, which calls for an urgent need for the development of rapid diagnostics that can accurately predict the susceptibility of the infecting strain [19]. For example, a lateral flow immunoassay for the extended-spectrum β-lactamase (ESBL) CTX-M has been shown to have excellent sensitivity and specificity for the rapid detection of CTX-M-producing *Enterobacterales* in blood culture [20]. Until the wide availability of rapid diagnostic tests, knowledge of the local, regional, and global distribution of antimicrobial resistance in *Salmonella* Typhi remains crucial for guiding empirical treatment and public health interventions.

In Hong Kong, typhoid fever has been a notifiable infectious disease since 1946. In the years between 2010 and 2015, the average annual number of cases reported to the Centre for Health Protection, Department of Health was 30 (range, 25 to 34) [21]. The annual number of cases decreased to 14–21 in 2016–2019, then further dropped to 12 in 2020 and 10 in 2021. However, two cases of locally acquired typhoid fever were reported within a two-week interval in late September to early October 2022. Both patients lived in the same region on Hong Kong Island, and their *Salmonella* Typhi isolates had similar antibiotic susceptibility patterns. Hence, we conducted a phylogenetic analysis of *Salmonella* Typhi isolates on Hong Kong Island from 2020 to 2022. Whole-genome sequencing (WGS), the typing of plasmids, and an analysis of antibiotic-resistance genes (ARGs) were performed to identify the dominant circulating strain and spread of ARGs

## 2. Materials and Methods

### 2.1. Isolation and Identification of Salmonella Typhi

This is a retrospective study on cases of typhoid fever diagnosed in Queen Mary Hospital (QMH) and Pamela Youde Nethersole Eastern Hospital (PYH), the two referral hospitals situated on Hong Kong Island, from 2020 to 2022. Because only blood culture isolates are routinely cryopreserved, cases of typhoid fever were identified by blood culture positivity using the Laboratory Information System. The clinical information, including patient demographics, underlying illness, symptomatology, travel history, laboratory test results, antimicrobial treatment, and disease outcomes were retrieved from the electronic patient record of the Hospital Authority. The *Salmonella* Typhi isolates stored in Microbank (Pro Lab Diagnostics Inc., Toronto, ON, Canada) were recovered and subcultured onto blood agar for subsequent testing. Bacterial identification on isolated colonies was performed using a combination of Matrix-Assisted Laser Desorption/Ionisation Time-of-Flight mass spectrometry (Bruker Daltonics, Bremen, Germany), which would identify the isolate to the genus level as “*Salmonella* spp.” Serotyping was performed by a slide agglutination test using Salmonella O, H and Vi antisera and the results were interpreted according to the White–Kauffmann–Le Minor Scheme [22]. Briefly, a dense bacterial suspension with saline was prepared from pure growth on non-selective agar. The bacterial suspension was mixed with polyvalent somatic (O), polyvalent flagella (H), and Vi antisera, respectively, and rocked gently. Agglutination was observed using indirect lighting over a dark background within one minute. Autoagglutination was checked by mixing the bacterial suspension with saline. If agglutination occurred with polyvalent O and polyvalent H antisera, further slide agglutination with specific O antigens and H antigens was performed. If serotyping with Vi antiserum was positive and O antisera was doubtful, the organism was heated at 100 °C for one hour, followed by a repeat agglutination test using O antisera. *Salmonella* Typhi was designated by positive reactions to O9, 12 (somatic), d (flagellar), and Vi (capsular).

### 2.2. Antimicrobial Susceptibility Testing

Antimicrobial susceptibility was determined using the disc diffusion method (for the β-lactams ampicillin, ceftriaxone, and meropenem, the 50S ribosomal protein synthesis inhibitor chloramphenicol, and the folate synthesis inhibitor co-trimoxazole) or minimum inhibitory concentration (MIC) test strips (Liofilchem, Teramo, Italy) (for the macrolide azithromycin and the fluoroquinolone ciprofloxacin) according to the standard set out by the Clinical and Laboratory Standards Institute (CLSI). The following inhibition zone diameters (mm) were used to interpret disc results (sensitive, intermediate, resistant): 10 µg of ampicillin (≥17, 14–16, ≤13), 30 µg of ceftriaxone (≥23, 20–22, ≤19), 30 µg of chloramphenicol (≥18, 13–17, ≤12), 1.25/23.75 µg of cotrimoxazole (≥16, 11–15, ≤10), and 10 µg of meropenem (≥23, 20–22, ≤19). The MIC (µg/mL) was interpreted as follows: azithromycin (sensitive ≤ 16, resistant ≥ 32) and ciprofloxacin (sensitive ≤ 0.06, intermediate 0.12–0.5, resistant ≥ 1). A double disc synergy test was used for the phenotypic detection of ESBL, in which ceftriaxone discs were placed 25 mm away from a 20/10 µg amoxicillin/clavulanate disc [20]. A clear extension of the edge of the inhibition zone towards the disc containing clavulanate was interpreted as positive for ESBL production. The antibiotic susceptibility test result was interpreted according to the breakpoints set out in the 2022 CLSI M100 document [23]. *Salmonella* Typhi strains that are non-susceptible to ampicillin, chloramphenicol, and co-trimoxazole were considered MDR, and strains with additional nonsusceptibility to fluoroquinolones and third-generation cephalosporins were considered to be extensively-drug resistant (XDR), in line with previous publications describing *Salmonella* Typhi resistance [16,24].

### 2.3. WGS Library Preparation and Sequencing

The *Salmonella* Typhi isolates were first inoculated in brain heart infusion broth without antibiotics at 35 °C for 18 h and genomic DNA was extracted using the Qiagen DNeasy Blood and Tissue kit (Qiagen, Hilden, Germany) according to the manufacturer’s instructions. The genomic DNA of each strain was first sheared with Covaris g-TUBE (Covaris Inc, Woburn, MA, USA) to an average DNA size of 10,000 bp. WGS was then performed using both Illumina and Nanopore sequencing. In brief, the DNA library for short-read sequencing was prepared using the Nextera DNA Prep Kit (Illumina Inc, San Diego, CA, USA) with Nextera DNA CD Indexes (Illumina Inc, San Diego, CA, USA). The libraries were sequenced on the Illumina iSeq100 platform using a 2 × 150 bp paired-end configuration. DNA libraries for long-read sequencing were prepared using the Nanopore Ligation Sequencing Kit (SQK-LSK109) and the Native Barcoding Kit (NBD-EXP104) (Oxford Nanopore Technologies, Oxford, UK) according to the manufacturer’s instructions. The libraries were sequenced using MinION flowcells vR9.4.1 (Oxford Nanopore Technologies, Oxford, UK) and run for 24–36 h. Basecalling and demultiplexing were performed using Guppy 6.1.2 with a super-accuracy basecalling model (dna_r9.4.1_450bps_sup) with a raw read accuracy of up to 98.3%. 

### 2.4. Genome Assembly and Annotation

Prior to genome assembly, the quality of both the short and long raw sequencing reads was first evaluated using the FastQC software (https://github.com/s-andrews/FastQC, accessed on 25 November 2022). The Illumina short reads were then trimmed with Trimmomatic v0.39 and the Nanopore long reads were trimmed by Porechop version 0.2.4. The trimmed reads proceeded to de novo assembly using Unicycler v0.4.9, as in our previous studies [25,26]. In the case of an incomplete genome generated from Unicycler, CANU v2.2 [27] or Flye v2.9.1 [28] was used to assemble the Nanopore reads and corrected with Pilon v1.24 [29]. Quality evaluations, fixations of the start position with *dnaA*, and annotations of genomes were performed as described in our previous studies [25,26].

### 2.5. Phylogenetic Analysis and Other Genome Analysis

All the complete genomes from the present study were aligned with *Salmonella* Typhi CT18 (GenBank accession AL513382) using Mauve to check the recombination events [30]. Genomes from the present study were assigned to genotypes using GenoTyphi v1.9.1 [18]. To further decipher the phylogenetic relationship of our strains, they were integrated with a global dataset of 711 genotype 3 genomes, which were retrieved from NCBI and previous publications (see Appendix A) [15,17,24,31,32,33,34,35,36,37,38]. All these type 3 genomes were aligned with *Salmonella* Typhi CT18 (GenBank accession AL513382) to call core SNPs using Snippy v4.6.0 with –mincov 10 and –minfrac 0.9 (https://github.com/tseeman/snippy, accessed on 25 November 2022). The CT18 reference genome belonged to genotype 3.2.1. The phylogenomic trees of type 3 and a genotype including our strains and 30 close strains were built using IQ-Tree v2.2.0 with a 1000-times bootstrap test and GTR + I + G model [39], and both trees were visualised using iTOL (https://itol.embl.de, accessed on 25 November 2022). The SNP matrix of 30 close strains with our genomes was generated using snp-dists with the alignment of core SNPs (https://github.com/tseemann/snp-dists, accessed on 25 November 2022). Two more phylogenomic trees of genotypes 2.3.4 and 4.3.1.1.P1 were constructed using the same pipeline above but with S7 and S3 genomes from the present study as references.

The typing of plasmids was performed using PlasmidFinder v2.1 [40] and fixed with the *repA* gene as a start position, as performed for the chromosomes above. The genomes were further queried for the presence of the linear plasmid, pBSSB1 which was reported to induce the flagellar phase change in *Salmonella* Typhi [5]. The complete plasmid pS3 of strain S3 from the present study was subjected to NCBI to search for closely related plasmids, which were aligned together with p60006 (GenBank accession LT906492.1) as a reference and presented in a circular map using BLAST Ring Image Generator (BRIG) v0.9.5 [41].

ARGs and point mutations in all strains were detected using ResFinder v4.0 [42], HMD-ARG v1.0 [43], and PointFinder [44]. CARD v3.2.5 was also run via Resistance Gene Identifier (RGI) v6.0.0 to confirm all ARGs and point mutations. All mobile genetic elements, including genomic islands (GIs), integrative and conjugative/mobilisable elements (ICE/IME), insertion elements (ISs), and prophages were identified as described previously [45]. 

The sequences of the isolates sequenced in this study were deposited in the GenBank database under Bioproject PRJNA910287. 

## 3. Results

### 3.1. Clinical Features of Cases with Salmonella Typhi Bacteraemia

We conducted a retrospective phylogenetic study of *Salmonella* Typhi isolates on Hong Kong Island between January 2020 and December 2022 to identify the dominant circulating strain, the relationship between cases, and the spread of ARGs. A total of seven cases (designated S1 to S7) with *Salmonella* Typhi bacteraemia were identified in QMH and PYH during the study period (Table 1). There were five females (71.4%). The median age was 51 years old (range, 4–73 years). Four (57%) had no underlying comorbidities. For the three patients with comorbid conditions, one (S1) was pregnant at 12 weeks gestation on admission and had rheumatoid arthritis on observation only, one (S3) had diabetes mellitus on oral hypoglycaemic agents and epilepsy, and one (S7) had gallstones and metastatic carcinoma of the gallbladder on palliative care. Patient S6 had symptom onset on the same day of arriving in Hong Kong from the Philippines and was designated as an imported case. The other six cases did not report any history of travel outside Hong Kong in the 12 months prior to symptom onset. All cases were reported to the Department of Health for epidemiological investigation. No epidemiological relationship including common exposure, contact history, or other links was identified among the cases. There was no direct contact between any two patients. The most common presenting symptom was fever, which was present in five patients (71%), followed by (in decreasing frequency) gastrointestinal symptoms (including abdominal pain, vomiting, diarrhoea, and nausea), respiratory symptoms (including cough and dyspnoea), and headache. Patient S3 developed an infected infrarenal aortic aneurysm requiring endovascular stenting and lumbar spondylodiscitis. All patients received β-lactam antibiotics, including meropenem in two patients, ceftriaxone in four patients, and piperacillin–tazobactam in one patient; four patients received azithromycin. Patient S7 succumbed shortly following admission before the blood culture turned positive. The other six patients made full recoveries. 

### 3.2. Antimicrobial Susceptibilities of the Recovered Salmonella Typhi Isolates

Seven *Salmonella* Typhi isolates were recovered from storage. All were isolated from blood culture. Stool was collected for culture in five patients, and two of them were culture-positive for *Salmonella* Typhi. The isolates from stool were not preserved and thus were not available for analysis. The antibiotic susceptibility patterns of the seven blood culture *Salmonella* Typhi isolates are shown in Table 1. Strain S3 exhibited an XDR phenotype with co-resistance to ampicillin, chloramphenicol, cotrimoxazole, ciprofloxacin (MIC 2 µg/mL), and ceftriaxone. The double disc synergy test for ESBL was positive. Strain S5 was ciprofloxacin intermediate (MIC 0.5 µg/mL) but susceptible to the other tested antibiotics. The other five isolates (S1, S2, S4, S6, S7) were susceptible to all the tested antibiotics. All isolates were susceptible to azithromycin (MIC 4–8 µg/mL) and meropenem. 

### 3.3. WGS of the Salmonella Typhi Isolates

All seven isolates were subjected to WGS. The chromosome size ranged between 4.776 and 4.900 Mb. The mauve alignment of the completed chromosomes revealed multiple recombination events relative to the reference genome CT18 (Figure 1). Of note, the overall genomic organisations of S1, S2, and S5 were identical. Additional rearrangements were observed in the other four genomes (S6, S4, S3, and S7). SNP-based genotyping using the GenoTyphi pipeline showed that S1, S2, and S4 to S6 were of genotype 3.2.2, strain S3 was of genotype 4.3.1.1.P1 (also known as H58 lineage), and strain S7 was of genotype 2.3.4 (Table 1). The strains were further compared with genomes of the same genotype deposited in GenBank using core genome SNP analysis. A comparison with over 700 genotype 3.2.2 genomes showed that the five genotype 3.2.2 strains in this study form a cluster with another 30 strains originating from Southeast Asia (Figure 2). The SNP differences for the 35 clustered strains were compared, showing that two of them, S1 and S2, were clonally related with zero SNP difference, compared with 15 to 94 SNPs among the other three strains (S4–S6). For genotype 4.3.1.1.P1, a total of 568 genomes were previously deposited in GenBank and they originated from Pakistan in 2016–2018. Phylogenetic analysis showed that the genotype 4.3.1.1.P1 strain S3 was genetically related to the cluster of 568 genomes with 38–42 SNP differences (Appendix A). For genotype 3.2.4, only a limited number of genomes (*n* = 16, from 1981 to 2017) have been deposited in GenBank. The genotype 3.2.4 strain S7 was most closely related to a strain from Vietnam (GenBank reference: ERR349376) in 1997 with 92 SNP, 6 insertion, and 8 deletion differences (Appendix A).

All seven strains had a single copy of the chromosomally integrated *aac(6′)-Iaa* gene. This ARG was not associated with any IS element. Additional ARGs were detected in strain S3 in an IncY plasmid pS3 and a chromosomally integrated composite transposon-like structure. Plasmid pS3 had a size of 84,390 bp and carried six ARGs including *sul2*, *aph(3”)-1b*, *aph(6)-I*, *bla*_TEM-1_, *bla*_CTX-M-15_, and *qnrS1*. Plasmid pS3 exhibited a high degree of identity (100.00% coverage and 99.99% identity) with the prototype p60006 plasmid (Figure 3), which is widespread among MDR *Salmonella* Typhi isolates. Another set of seven ARGs (*dfrA7*, *qacE*, *sul1*, *bla*_TEM-1_, *aph(6)-1d*, *aph(3”)-1b*, and *sul2*) were located in a chromosomally integrated composite transposon-like structure. S3 is the only strain with a plasmid. The other six strains did not contain plasmids. In all the strains, no pBSSB1-like linear plasmid was detected. In strains S3 and S5, S83F substitution in the *gyrA* gene was detected. No quinolone resistance-determining region (QRDR) mutation was detected in the other five strains. 

## 4. Discussion

In this study, we performed a phylogenetic analysis of seven *Salmonella* Typhi isolates from Hong Kong Island over a three-year period, including six local and one imported strains. The predominance of genotype 3.2.2 and clonal transmission between the two index cases, S1 and S2, were demonstrated. In addition, a local strain with XDR phenotype belonging to the H58 lineage was found, suggesting that this globally dominant XDR lineage has spread to Hong Kong with resulting local transmission. To our knowledge, this is the first genomic study on the *Salmonella* Typhi genotypes in Hong Kong.

Antibiotic-resistant typhoid fever has long been a public health concern in Southern Asia. A clonal expansion of the H58 lineage (genotype 4.3.1) has been observed since the 2000s, with most fluoroquinolone-resistant strains possessing one of six nonsynonymous mutations at codon 83 and 87 of *gyrA* [13]. Currently, H58 (genotype 4.3.1) is subdivided into three lineages by intra-H58 SNPs. H58 lineages I (genotype 4.3.1.1) and II (genotype 4.3.1.2) were first defined in a study on *Salmonella* Typhi isolates from Nepalese children [46]. H58 lineage I is further classified into different sublineages following their introduction into different geographic areas and subsequent outbreaks [18]. A recent surveillance study carried out in Kenya found that all H58 lineage I isolates belong to the sublineage EA1 (East Africa 1) with genotype designation 4.3.1.1.EA1, and H58 lineage II isolates belong to two distinct sublineages designated as EA2 (East Africa 2, genotype 4.3.1.2.EA2) and EA3 (East Africa 3, genotype 4.3.1.2.EA3). All three East African H58 genotypes had high rates of MDR, with the MDR phenotype in most EA1 and all EA2 strains being associated with the plasmid sequence type 6 (PST6)-IncHI1 plasmid, carrying a suite of ARGs, while in EA3 it was exclusively associated with the chromosomal insertion of multiple ARGs [47]. H58 lineage III (genotype 4.3.1.3) is a cluster of isolates mostly from Bangladesh. A sublineage of genotype 4.4.1.3 with fluoroquinolone resistance has been designated as genotype 4.3.1.3.Bdg [18]. 

In November 2016, Pakistan experienced a multi-province outbreak of an XDR *Salmonella* Typhi clone within the H58 lineage that harbours additional resistance genes, including the *bla*_CTX-M-15_ ESBL and the *qnrS* fluoroquinolone resistance gene encoded on an IncY plasmid likely acquired from other enteric bacteria [16,48]. Subsequent WGS revealed that all isolates in this outbreak were sequence type (ST)1 and were identified as genotype 4.3.1.1.P1 under the GenoTyphi scheme [48]. Within 3 years of its emergence, genotype 4.3.1.1.P1 has become the dominant genotype in Pakistan [24]. The World Health Organisation estimated that, between 2016 and 2018, more than 5000 cases occurred in Sindh province, Pakistan [49]. Imported cases of XDR *Salmonella* Typhi infection in patients with a history of travel to Pakistan have been reported in the United States [50], Canada [51], the United Kingdom [52], Beijing [53], and Australia [54], although genotyping was not always performed in the cases to confirm that the isolates belong to genotype 4.3.1.1.P1. In strain S3, which demonstrates a similar XDR phenotype, an IncY plasmid with a high degree of sequence identity to the prototype p6006 plasmid was found that carried ARGs including *bla*_CTX-M-15_ and *qnrS1.* This is the first report of XDR *Salmonella* Typhi of genotype 4.3.1.1.P1 in Hong Kong. 

In this study, we defined the XDR phenotype as non-susceptible to fluoroquinolones and third-generation cephalosporins in addition to the three first-line antimicrobials, the same as the definitions used in the initial report on XDR *Salmonella* Typhi from Pakistan and subsequent publications [16,24,53,54,55]. This is different from the interim standard definitions proposed by Magiorakos et al., in which XDR is defined as non-susceptible to all but two antibiotic classes recommended for treatment [56]. However, it should be pointed out that the standard definition is only applicable to Enterobacteriaceae other than *Salmonella* and *Shigella* [56]. Hence, it is not applicable to *Salmonella* Typhi. 

In addition, we found that one imported and four local isolates were of genotype 3.2.2. All five isolates were susceptible to the three first-line antimicrobials ampicillin, chloramphenicol, and co-trimoxazole. Only isolate S5 demonstrated non-susceptibility to ciprofloxacin, consistent with the detection of S83F substitution in the *gyrA* gene. Similar phenomena of the locally restricted spread of antibiotic-sensitive clones of *Salmonella* Typhi have been observed in other regions [57]. Using data from the WGS of nearly 2000 isolates from more than 60 countries, Wong et al. concluded that with the exception of the rapidly disseminating H58 lineage, the global *Salmonella* Typhi population is highly structured and includes many genotypes that display geographical restriction [17]. A recent genomic epidemiology study also revealed that genotype 3.2.2 was among four genotypes that were dominant in South Asia [24]. However, considering the frequent population movement between Hong Kong and Southern Asian countries, the risk of the importation of *Salmonella* Typhi strains with additional MDR or XDR phenotypes that may displace the locally predominant genotype 3.2.2 *Salmonella* Typhi population is substantial. 

In the face of the global dissemination of MDR *Salmonella* Typhi, third-generation cephalosporins such as ceftriaxone have become the recommended treatment of choice for most patients with severe or complicated typhoid fever without a recent history of travel to areas with a high prevalence of XDR strains [58]. Patients with recent travel to areas where XDR *Salmonella* Typhi is common, such as Pakistan and Iraq [59], are recommended to receive a carbapenem such as meropenem as empirical treatment. Recently, carbapenem-resistant *Salmonella* Typhi isolates harbouring the carbapenemase genes *bla*_VIM_ and *bla*_GES_ have been reported in Pakistan [60], although susceptibility to other antimicrobials and the genotypes were not specified. The emergence of carbapenem resistance in *Salmonella* Typhi represents an imminent public health threat, as treatment options would be further limited. 

There are several limitations to this study. First, we only included isolates from Hong Kong Island over a three-year period. A territory-wide genomic surveillance study involving other regions of Hong Kong will be needed to analyse the regional distribution of *Salmonella* Typhi genotypes. Second, there was only one imported case from 2020 to 2022 in this study due to greatly reduced travelling during the Coronavirus Disease 2019 (COVID-19) pandemic. A further study spanning a longer interval and the testing of more isolates may reveal additional introduction events and more genotypes. Third, we only included isolates from blood culture since bacterial isolates from other sites were not cryopreserved. However, blood culture is only positive in 60–80% of patients and the sensitivity decreases after the first week of illness [2]. Furthermore, *Salmonella* Typhi clones may also be introduced and spread through asymptomatic carriage [47]. A prospective study including *Salmonella* Typhi isolates from all sites would provide a more comprehensive picture.

## 5. Conclusions

In conclusion, we demonstrated a predominance of antibiotic-susceptible genotype 3.3.3 *Salmonella* Typhi on Hong Kong Island and clonal spread between cases. Moreover, the first local case of XDR *Salmonella* Typhi of genotype 4.3.1.1.P1 was detected. Although the majority of local strains had a low risk of antibiotic resistance, the introduction and local spread of XDR clones remain a public health threat.

## Figures and Tables

**Figure 1 microorganisms-11-00667-f001:**
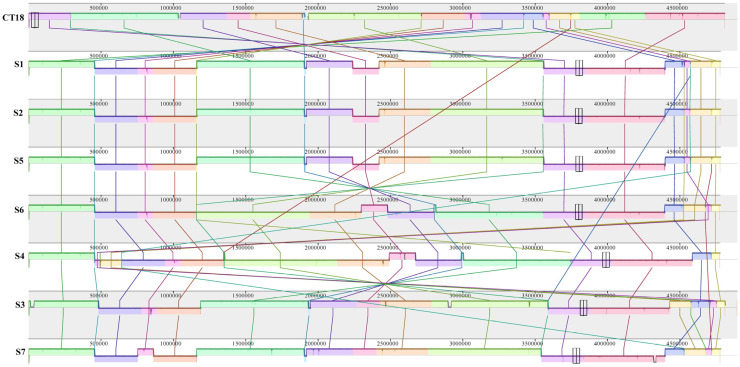
Mauve alignment of S1 to S7 genomes with the *Salmonella* Typhi reference CT18. Each coloured block (called locally collinear blocks) represents a region of homologous sequence that aligns to part of another genome and is free from internal rearrangement. Color lines indicate which blocks in each genome are homologous.

**Figure 2 microorganisms-11-00667-f002:**
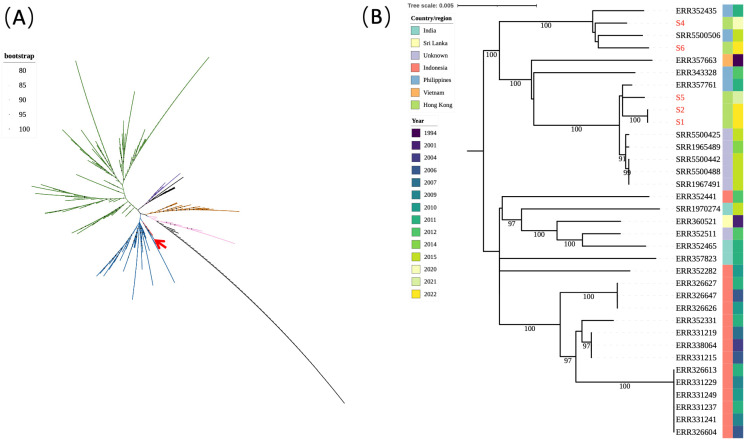
Phylogenetic analysis of 35 *Salmonella* Typhi strains of genotype 3.2.2. (**A**) An unrooted maximum likelihood phylogenetic tree of 716 *Salmonella* Typhi isolates of clade 3. The red arrow indicates the branch with 35 isolates, including the 5 isolates from the present study. (**B**) A higher resolution diagram of the branch from panel (**A**). The five genotype 3.2.2 isolates from Hong Kong (S1–S2, S4–S6) are indicated in red font.

**Figure 3 microorganisms-11-00667-f003:**
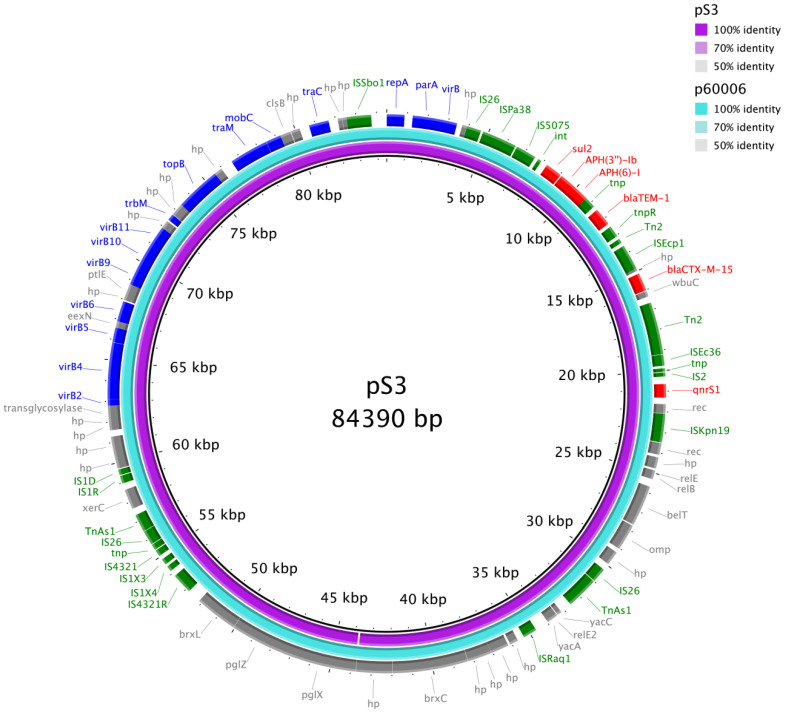
Alignment of the plasmid pS3 in this study with the reference plasmid p60006 using BLAST Ring Image Generator (BRIG).

**Table 1 microorganisms-11-00667-t001:** Clinical and microbiological details of cases with *Salmonella* Typhi bacteraemia.

Characteristic	Case ^a, b^						
	S1	S2	S3	S4	S5	S6	S7
Sex/age (years)	Female/39	Female/23	Male/54	Male/51	Female/4	Female/51	Female/73
Date of symptom onset	Oct 2022	Sep 2022	Feb 2022	Aug 2020	Jul 2021	Jun 2022	May 2021
Presenting symptoms	Fever, chills, rigour, nausea, vomiting, diarrhoea, abdominal pain	Fever, chills, headache	Abdominal and lower back pain, nausea, vomiting	Fever, chills, cough, dyspnoea, abdominal pain, diarrhoea, vomiting	Fever, fatigue	Fever, cough, vomiting	Cough, dyspnoea, abdominal pain
Complications	Nil	Nil	Infected abdominal aortic aneurysm, lumbar spondylodiscitis	Nil	Nil	Nil	Renal failure Coagulopathy Mortality
Travel outside HK in preceding 12 months	No	No	No	No	No	Yes (Philippines)	No
Culture positive site	Blood	Blood	Blood, stool	Blood	Blood, stool	Blood	Blood
Susceptibility							
Ampicillin	S	S	R [*bla*_TEM-1B_, *bla*_CTX-M-15_]	S	S	S	S
Azithromycin	S (8)	S (4)	S (8)	S (4)	S (4)	S (4)	S (8)
Ceftriaxone	S	S	R [*bla*_CTX-M-15_]	S	S	S	S
Chloramphenicol	S	S	R [*catA1*]	S	S	S	S
Ciprofloxacin	S (0.012)	S (0.012)	R (2) [*gyrA* S83F, *qnrS1*]	S (0.012)	I (0.5) [*gyrA* S83F]	S (0.06)	S (0.06)
Co-trimoxazole	S	S	R [*sul1*, *sul2*, *dfrA7*]	S	S	S	S
Meropenem	S	S	S	S	S	S	S
Genotype	3.2.2	3.2.2	4.3.1.1.P1	3.2.2	3.2.2	3.2.2	2.3.4

S—susceptible; I—intermediate; R—resistant. ^a^ MIC value is given inside parentheses ( ). Resistance determinant is given inside brackets [ ]. ^b^ 2022 CLSI MIC interpretative criteria (µg/mL): azithromycin, S ≤ 16, R ≥ 32; ciprofloxacin, S ≤ 0.06, I 0.12–0.5, R ≥ 1.

## Data Availability

All data has been included in this manuscript and the Appendix A.

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
