# Peer review of "Genomic Investigation of Salmonella Typhi in Hong Kong Revealing the Predominance of Genotype 3.2.2 and the First Case of an Extensively Drug-Resistant H58 Genotype"

_microorganisms, 2023, doi:10.3390/microorganisms11030667_

Round 1
Reviewer 1 Report
Good report overall very well written and data is well analysed - It is acceptable as a brief report and authors addressed the study's limitations in detail - the main comment that the WGS reads are not available under project PRJNA910287 and data should be released.
Author Response
We have asked GenBank to release the sequences.
Reviewer 2 Report
Comments to authors:
-The current study is interesting; however, the authors should address the following comments to improve the quality of the manuscript:
-The manuscript should be revised for English editing and grammar mistakes.
- Please write the scientific names of bacterial pathogens and genes in the correct form all over the manuscript and the references section.
Title:
I think the work would benefit from the title that contains the main conclusion of the study (should be derived from the conclusion). Please modify the title.
Abstract:
- The abstract must illustrate the used methods and the most prevalent results (give more hints about methods and results). Besides, rephrase the aim of the work and the main conclusion of your findings.
-A graphical abstract is recommended (If possible).
- Add the full expression before the abbreviations.
-Introduction: (it needs to be more informative):
-Give a hint about the virulence factors, the mechanism of disease occurrence, and infecions caused by Salmonella Typhi.
- The authors should illustrate the public health importance concerning the emergence of virulent multidrug-resistant (MDR) bacterial pathogens that reflect the necessity of application rapid specific diagnostic tools;
Authors could add the following paragraph:
Multidrug resistance has been increased all over the world that is considered a public health threat. Several recent investigations reported the emergence of virulent multidrug-resistant bacterial pathogens from different origins that increase the necessity of the proper use of antibiotics as well as the application of rapid accurate diganostic tools for screening of the emerging virulent MDR strains. You are advised to cite the following valuable studies:
1. PMID: 35971557
2. PMID: 36365013
3. PMID: 34445951
-Rephrase the aim of the work to be clear and better sound.
Material and methods:
- Support all methods with updated specific references.
• Add the company, city, and country of the used chemicals and reagents.
- Add this subtitle: Isolation and identification of Salmonella Typhi:
Discuss in detail the methods of isolation and identification of Salmonella Typhi. Besides, specific references should be added.
- Disscuss in details serotyping of the recovered Salmonella Typhi isolates.
-Antimucrobial susceptibility testing:
-Please, explain in detail.
•Add the names of all tested antibiotics and the antimicrobial classes.
•The authors are advised to classify the tested isolates to MDR , XDR, and PDR as described by Magiorakos et al.
Magiorakos AP, Srinivasan A, Carey RB, Carmeli Y, Falagas ME, Giske CG, et al. Multidrug-resistant, extensively drug-resistant and pandrug-resistant bacteria: An international expert proposal for interim standard definitions for acquired resistance. Clin Microbiol Infect. 2012; 18:268–81. doi:10.1111/j.1469-0691.2011.03570.x.
- To increase the impact of the present study, the detection of virulence and antimicrobial resistance genes in the recovered S. Typhi should be performed. Afterwards, the correlation between phenotypic and genotypic multidrug resistance should be performed.
-Data analysis: Add more details about the used software.
-Results: (Good Presentation)
- Please add a starting paragraph to the results section to briefly introduce the topic, your goals and
hypothesis and a short summary of what you did in this work.
-Add this subtitle: Phenotypic characteristics of the recovered Salmonella Typhi strains:
• Illustrate in detail the phenotypic characteristics of the recovered isolates.
-Antimicrobial susceptibility testing:
• -Illustrate in a new table the occurrence of MDR (Multidrug resistance) among the recovered isolates as the following (illustrate the names of the antimicrobial classes and different antibiotics):
No. of strains % Type of resistance
R, MDR, and XDR Phenotypic multidrug resistance
(Antimicrobial classes and different antibiotics). The antibiotic-resistance genes
-To increase the impact of the present study, the detection of virulence and antimicrobial resistance genes in the recovered S. Typhi should be performed. Afterwards, the correlation between phenotypic and genotypic multidrug resistance should be performed.
-Increase the resolution of all figures (must be 600 dpi).
-Discussion:
The authors are advised to illustrate the real impact of their findings without repetition of results.
- Please illustrate the mechanism of action of different virulence determinants of Salmonella Typyhi.
- Please illustrate the mechanism of antimicrobial resistance in Salmonella Typhi.
-Conclusion
- Should be rephrased to be sounded. A real conclusion should focus on the question or claim you articulated in your study, which resolution has been the main objective of your paper?
Author Response
The current study is interesting; however, the authors should address the following comments to improve the quality of the manuscript:
- The manuscript should be revised for English editing and grammar mistakes.
Response: We have edited the manuscript and corrected the grammatical mistakes identified.
- Please write the scientific names of bacterial pathogens and genes in the correct form all over the manuscript and the references section.
Response: We agree that although the abbreviated name S. Typhi is widely used in the literature, it does not strictly follow the CDC designation [1, 2]. Hence, we have updated the pathogen name to Salmonella enterica subsp. enterica serotype Typhi when it first appeared and replaced all S. Typhi with Salmonella Typhi, in line with the nomenclature rules. We have also checked the names of genes to ensure that they are the commonly used names in the scientific literature. The pathogen and gene names in the reference section are from the original publications.
References:
- Prevention, C.f.D.C.a. Scientific Nomenclature. March 02, 2022 [cited 2022 February 17]; Available from: https://wwwnc.cdc.gov/eid/page/scientific-nomenclature?pastIssue=/eid/articles/issue/21/7/table-of-contents.
- Brenner, F.W., et al., Salmonella nomenclature. J Clin Microbiol, 2000. 38(7): p. 2465-7.
Title:
I think the work would benefit from the title that contains the main conclusion of the study (should be derived from the conclusion). Please modify the title
Response: We have revised the title to ‘Genomic investigation of Salmonella Typhi in Hong Kong revealing the predominance of genotype 3.2.2 and the first case of extensively drug-resistant H58 genotype’.
Abstract:
- The abstract must illustrate the used methods and the most prevalent results (give more hints about methods and results). Besides, rephrase the aim of the work and the main conclusion of your findings.
Response: Thank you for the suggestion to improve it. The abstract has been revised according to these suggestions.
- A graphical abstract is recommended (If possible).
Response: We decide not to include a graphic abstract.
- Add the full expression before the abbreviations.
Response: We have added full expression when the abbreviations first appeared in the manuscript.
Introduction: (it needs to be more informative):
- Give a hint about the virulence factors, the mechanism of disease occurrence, and infections caused by Salmonella
Response: The Introduction section has been extensively rewritten. The suggested areas including virulence factors, mechanism of disease and infections have been added in the revision.
- The authors should illustrate the public health importance concerning the emergence of virulent multidrug-resistant (MDR) bacterial pathogens that reflect the necessity of application rapid specific diagnostic tools.
Response: This has been revised according to the suggestion. In the Introduction section, we added one paragraph to describe the necessity for rapid diagnostic tests.
Authors could add the following paragraph:
Multidrug resistance has been increased all over the world that is considered a public health threat. Several recent investigations reported the emergence of virulent multidrug-resistant bacterial pathogens from different origins that increase the necessity of the proper use of antibiotics as well as the application of rapid accurate diagnostic tools for screening of the emerging virulent MDR strains. You are advised to cite the following valuable studies:
- PMID: 35971557
- PMID: 36365013
- PMID: 34445951
Response: We agree fully with the reviewer that the emergence of multidrug resistance is a public health threat, as we have stated in the Introduction part and reiterated in the discussion. We have studied the references suggested by this reviewer. While we agree that those references contained useful information, the content is unrelated to Salmonella Typhi. Hence, they were not added.
- Algammal AM, Hashem HR, Al-Otaibi AS et al. Emerging MDR-Mycobacterium avium subsp. avium in house-reared domestic birds as the first report in Egypt. BMC Microbiol 2021; 21, 237.
- Algammal AM, Ibrahim RA, Alfifi KJ et al. A First Report of Molecular Typing, Virulence Traits, and Phenotypic and Genotypic Resistance Patterns of Newly Emerging XDR and MDR Aeromonas veronii in Mugil seheli. Pathogens 2022; 11.
- Algammal AM, Abo Hashem ME, Alfifi KJ et al. Sequence Analysis, Antibiogram Profile, Virulence and Antibiotic Resistance Genes of XDR and MDR Gallibacterium anatis Isolated from Layer Chickens in Egypt. Infect Drug Resist 2022; 15, 4321-34.
- Rephrase the aim of the work to be clear and better sound.
Response: We have rephrased the aim of the work accordingly.
Material and methods:
- Support all methods with updated specific references.
- Add the company, city, and country of the used chemicals and reagents.
Response: Suitable references have been cited for all methods. Details on the chemicals and reagents have been added.
- Add this subtitle: Isolation and identification of Salmonella Typhi.
Response: The subtitle has been added.
- Discuss in detail the methods of isolation and identification of Salmonella Besides, specific references should be added.
Response: The isolation and identification have been described in detail with references.
- Discuss in details serotyping of the recovered Salmonella Typhi isolates.
Response: The serotyping method has been described in detail.
Antimicrobial susceptibility testing:
- Please, explain in detail.
- Add the names of all tested antibiotics and the antimicrobial classes.
Response: This has been added accordingly.
- The authors are advised to classify the tested isolates to MDR , XDR, and PDR as described by Magiorakos et al..
Magiorakos AP, Srinivasan A, Carey RB, Carmeli Y, Falagas ME, Giske CG, et al. Multidrug-resistant, extensively drug-resistant and pandrug-resistant bacteria: An international expert proposal for interim standard definitions for acquired resistance. Clin Microbiol Infect. 2012; 18:268–81. doi:10.1111/j.1469-0691.2011.03570.x.
Response: In the revised manuscript, we explained the reason for the MDR and XDR definitions in this study. The definitions proposed by Magiorakos et al have been added to the Discussion section. While those definitions have been widely adopted by many investigators, they are not applicable to Salmonella and Shigella. In the abstract https://pubmed.ncbi.nlm.nih.gov/21793988/ “A group of international experts came together through a joint initiative by the European Centre for Disease Prevention and Control (ECDC) and the Centers for Disease Control and Prevention (CDC), to create a standardized international terminology with which to describe acquired resistance profiles in Staphylococcus aureus, Enterococcus spp., Enterobacteriaceae (other than Salmonella and Shigella), Pseudomonas aeruginosa and Acinetobacter spp., all bacteria often responsible for healthcare-associated infections and prone to multidrug resistance.”
- To increase the impact of the present study, the detection of virulence and antimicrobial resistance genes in the recovered S. Typhi should be performed. Afterwards, the correlation between phenotypic and genotypic multidrug resistance should be performed.
Response: We performed analysis of the plasmids and antibiotic-resistance genes as described in 2.5. Phylogenetic analysis and other genome analysis. The results and correlation were detailed in the Result and Discussion section respectively.
- Data analysis: Add more details about the used software.
Response: We have revised the method section. Suitable reference have bene cited and elaboration provided for all bioinformatics tools.
Results: (Good Presentation)
- Please add a starting paragraph to the results section to briefly introduce the topic, your goals and hypothesis and a short summary of what you did in this work.
Response: We have added an introduction part in the first paragraph of Results to describe the topic, the goals, and a short summary, of the work.
- Add this subtitle: Phenotypic characteristics of the recovered Salmonella Typhi strains:
Response: The subtitle has been added.
- Illustrate in detail the phenotypic characteristics of the recovered isolates.
Response: We have described the antibiotic susceptibility pattern in detail both in section 3.2 and in Table 1. Other phenotypic characteristics, such as the colony morphologies and biochemical reactions etc., are not the primary focus of this article thus not described.
- Antimicrobial susceptibility testing:
- Illustrate in a new table the occurrence of MDR (Multidrug resistance) among the recovered isolates as the following (illustrate the names of the antimicrobial classes and different antibiotics):
No. of strains % Type of resistance
R, MDR, and XDR Phenotypic multidrug resistance
The antibiotic-resistance genes
Response: In Table 1, we have included the antibiotics for defining the organisms into MDR (ampicillin, chloramphenicol, co-trimoxazole) and XDR (ciprofloxacin and ceftriaxone) phenotypes. We arranged the antibiotics in alphabetical order for easy referencing. The definitions have been included in the Methods section. Because of the small number of isolates, we don’t feel the need to include a separate % column. In addition, because some of the resistance genes are carried in the same plasmid, separating the calculations of percentage of resistance under each antibiotic does not reflect the underlying genetic structure. We did not repeat the class of antibiotics in the table because it has been elaborated in the Methods section.
- To increase the impact of the present study, the detection of virulence and antimicrobial resistance genes in the recovered S. Typhi should be performed. Afterwards, the correlation between phenotypic and genotypic multidrug resistance should be performed.
Response: We performed analysis of the plasmids and antibiotic-resistance genes as described in 2.5. Phylogenetic analysis and other genome analysis. The results and correlation were detailed in the Result and Discussion section respectively. We did not perform separate analysis of the virulence genes because they are not the focus of the manuscript.
- Increase the resolution of all figures (must be 600 dpi).
Response: We have increased the resolution of all figures to above the required value. All figures in the WORD file have been replaced. We also upload the Figures as separate files.
Discussion:
The authors are advised to illustrate the real impact of their findings without repetition of results.
- Please illustrate the mechanism of action of different virulence determinants of Salmonella
Response: We have revised the first paragraph of the discussion to illustrate the significance of the findings without repeating the results. The virulence factor of Salmonella Typhi is not the focus of this manuscript thus not included in the discussion.
- Please illustrate the mechanism of antimicrobial resistance in Salmonella
Response: This has been added in the Discussion section.
Conclusion
- Should be rephrased to be sounded. A real conclusion should focus on the question or claim you articulated in your study, which resolution has been the main objective of your paper?
Response: Thank you for the suggestion. The conclusion has been rewritten accordingly.
Round 2
Reviewer 2 Report
The authors have carried out significant changes to the manuscript. They have addressed most of the suggested corrections and comments. Really, it's an interesting study that has a significant impact. Now, the manuscript could be accepted
Author Response
Author did not include the pS3 plasmid sequence assembly for sample S3 in GenBank submission; since they performed both short read and long read sequencing plasmid assembly will be closed and circular, please explain?
Response
Please notice that we have submitted plasmid of S3 with chromosome with accession number CP118537(chromosome) and CP118538(pS3).
Did author submit FastQ files via SRA submission? if not, it is highly recommended to submit the original FastQ files of seven samples via SRA submission.
Response
The sequence files including fastq have been uploaded and released by GenBank
https://www.ncbi.nlm.nih.gov/search/all/?term=PRJNA910287
